# Liberation Theologies and Their Future: Rethinking Categories and Popular Participation in Liberation

Joerg Rieger [1,*] and Priscila Silva [2]

1   Graduate Department of Religion and Divinity School, Vanderbilt University, Nashville, TN 37240, USA
2   Religious Studies Department, Methodist University, São Paulo 09640-000, Brazil; priscilapromotora@yahoo.com.br
*   Correspondence: j.rieger@vanderbilt.edu

**Abstract:** The first generation of Latin American liberation theologies was marked by the methodological status of the preferential option for the poor. In the following generations, this commitment was further developed in the struggle for a new way of doing theology, even more connected to material life, and disciplines such as history and economics were added. With this, the organizational structures of life in society started to be discussed in more critical, systemic, and prophetic ways. Especially thinking of the Latin American and US contexts, the production of theology derived from this intersectionality seeks not only to highlight and analyze the economic structures that cause exploitation (class), inequalities (gender and sexuality), and racism, but to identify how religion undergirds solidarity movements. The method applied to discuss these themes is bibliographical research. As a broad conclusion, this article indicates that future liberation theologies should discuss what the multiple victims of capitalism (always the majority of the population, never merely a minority) do in order to survive, related to the alternatives they create; discuss solidarity as the foundation that opposes social evil; and discuss the illusions of individualism that cover up both existing relationships of exploitation as well as solidarity.

**Keywords:** liberation theology; economics; solidarity; inequalities; Latin America; United States

## 1. Introduction

The liberation theologies born in the context of Latin American dictatorial systemic violence between the 1950s and 1970s continue to be debated, produced, and criticized. After the prominent theological developments of the nineteenth century, including the emergence of fundamentalism and liberalism in Europe, which influence the study of theology and doctrinal discussions to this day, the twentieth century was marked in Christian history by the birth of the theological imagination of Catholics and Protestants of the "third world". This includes the socioeconomic categories of the time, themes that were not intended to be systematic but methodological-ethical. These themes were not abstract reflections about God but addressed divine participation in the pains of the world, which influence our own participation and ethical-spiritual indignation.

Assuming not only the methodological theoretical conventions of disciplines such as philosophy, but including history as a foundation of production and bringing together theology and social sciences, liberation theologies and their forerunners were accused, among many things, of being traitors and rioters.[1] These accusations resembled earlier accusations of the Jesus movement as leading to sedition. We might argue that these theologies took the forms of protest against some empire, understood as a totality that generates death. All opposition to this new way of producing theology, more connected to the daily life of communities, demonstrated not only that the church was willing to surrender its own members to the accusatory powers—as happened with Rubem Alves, for example—but that a new historical period had begun. In the midst of economic dominance,

the globalization of production and the culture of consumption, in addition to the political-ideological persecution of people through dictatorships, liberation theologies sought to debate and organize society from different horizons, in contradiction to what the liberal economic universe preached. In this sense, liberation theologies not only collided with a certain theological tradition and conservative religious institutions of European molds of humanity, but vehemently opposed poverty-causing policies from the outset, denouncing oppressive forms of organized life in society.

In this way, liberation theologies, often considered heretical for their demands to break oppression, marked the history of the Roman Catholic Church and revisited Protestant perspectives of how to interpret biblical texts in the light of the real life of real people: the poor of Latin and Central America. Although commonly treated as a diamond cut from the intellectual work of Gustavo Gutiérrez, a Peruvian and Dominican friar, liberation theology was a movement, a process that had been underway since the 1950s through various ecumenical student collectives, which addressed the need to promote actions of a revolutionary nature on a continent marked by exploitation and the hardening of doctrine that made people and institutions indifferent to suffering in material life. Vatican II, an institutional seed of liberation theology, which took place between 1962 and 1965, was not a naive movement, but a takeover of the church's position in the face of the visible lack of meaning of collective faith in a new world, with new challenges for the faithful. Updating the church, *aggiornare*, bringing it into the open, was the proposal of John XXIII, pope at the time, to gather the church amid the cries for renewal and even repentance in light of the hegemonic-imperial Catholic faith.

A little more than 50 years later, in the face of constant discussions whether it still makes sense to talk about this topic, writing about the theologies of liberation (assuming their plurality) is a way of indicating that yes, this ongoing theological work is not only a memory but keeps germinating in various formats around the world, in places where processes and systems of domination, exploitation, and oppression persist. With the establishment of neoliberalism[2] and its global dimension, the vicissitudes that the young revolutionary Christians of the first generations discussed have become even worse and cruel, and we experience attempts to naturalize the indifferent, indiscriminate, political, economic, and theological use of individualism that marks Western societies. This can be seen, for instance, in how immigrants around the world are treated, as well as the working majorities of our countries who are forced to endure very unfavorable conditions.

Liberation theologies, considering these new times, embody the constant need for Christian theology to rethink its statutes and praxis in a world marked not only by the majority of population seeking ways to survive (historically classified as the poor), but by pain, social suffering, and exploitation that has been transformed into virtue, into a culture of "hard work," and into the myths of progress and meritocracy. Currently, the experience of neoliberal exploitation is not limited to Latin America but invades even hegemonic countries such as the United States, making everyday life a challenge, a struggle for survival on the backdrop of the accumulation of the few and the lack of the majority. In addition to people being forced into a production system without pause, nature itself cries out for freedom, for rest. Therefore, everyday life is a problem that needs to be addressed by theology and theologians. Producing liberation theologies today means, first of all, debating and proposing more just and solidary alternatives for collective life, whether in the North or South.

In view of these issues, this article will be divided into three sections that highlight some aspects of liberation theologies in the Americas at present, especially from the third generation onwards, identified by the critical intersection between theology and economics. Section 1 will engage material life through the lenses of liberation theologies, briefly reviewing capitalism as/and the study of religion. Section 2 will address idolatry, victims, sacrifice, and other intersectionalities as well as new categories of approach to an old oppression system. Section 3 deals with experiences of liberation in daily life in light of the development of "solidarity circles" in the United States. Finally, it is important to highlight

that this text is a collaboration of two theologians deeply influenced by liberation theologies in two completely different contexts. A Brazilian Baptist theologian, female, born and raised in a favela in Rio de Janeiro, and a German-American Methodist theologian, male, born and raised in Germany, and a professor in the United States. Both are marked, each in their own way, not only by the method of liberation theologies, but by their living fruits and results.

## 2. Development

### 2.1. The Material Life through the Lens of Liberation Theologies: Brief Review of Capitalism as/and the Study of Religion

Classical theology, the heritage of the fathers of the Church, and an overlay of centuries-old theological and philosophical schools, were shaped above all by a series of dichotomies that seemed irreconcilable: sacred and profane, flesh and spirit, spiritual life and material life, etc. In the period of birth of liberation theologies, this structure of thinking and performing the Christian faith in the world reproduced, in a certain sense, the landmarks of so-called secularization as well as the stratification of knowledge and life. On the one hand was faith, on the other reason; on one side was God, on the other science; on one side were spiritual things, on the other material things, and so on. Christian theology, in this scheme, was synonymous with knowing things about God. The consequence of this approach was that the practice of the Christian faith was associated with the search for a life outside of one's own life, untouched by the escalation of violence taking the forms of racism, patriarchy, wars, and misery in the post-war period. This amounted to lingering in a spiritual limbo within real life and to waiting for a heaven inside a desolate earth.

In an attempt to point out the need for Christian faith to look at real life, to position itself in the face of the widespread oppression that was at stake, Gustavo Gutiérrez indicates in his classic work *A Theology of Liberation*:

> This work seeks a reflection, from the Gospel and the experiences of men and women committed to the process of liberation *in this subcontinent of oppression and exploitation* that is Latin America. Theological reflection is born of this *shared experience* in the *effort to abolish the current situation of injustice and the construction of a different, freer and more humane society*. The route of liberating commitment was undertaken by many in Latin America. (Gutierrez [1971] 2000, p. 51, our translation)

Gutierrez points to the oppression and exploitation of Latin America as the result of historical processes that become experiences and are not just ideas. He also indicates that the ideal would be abolition, not just improvement of a particular situation. What Gutiérrez's liberation theology was protesting was a theology of development that assumed increased inclusion into an expanding system of global capitalism would be possible for Latin American countries and elsewhere. In addition, it is worth remembering that between the 1960s and 1980s, almost all Latin American countries went through periods of military dictatorship. The Argentinean,[3] Chilean, and Brazilian dictatorships stand out for significant levels of violence and repression, as well as for their economic extremes: massive industrialization was introduced at the price of subsequent mass poverty. The expected tide of growth turned out to be nothing more than a new wave of inequality that continues today.

What we want to indicate in reference to this text of Gutiérrez, often considered the father of father of liberation theology, is that it not only attempted to understand the attributes of God and the central elements of the Christian faith but to undo the dichotomies that distanced theological labor and the consequent practice of faith from the harsh reality of the life of the masses. Post World War II, Latin America had become a field of economic experiments (cheap labor, raw material exploitation, etc.), mainly from the United States, and liberation theology proposed a method—"see, judge, act"—that tried to reconcile faith and its meaning in a context of "open veins," as Eduardo Galeano used to say.[4]

Thus, the first and second generations of liberation theologians were focused on defending the choice for the poor as a theological priority, on structuring methodological

principles and, as a consequence, on developing the first contents of a soteriology, ecclesiology, and pneumatology from the perspective of liberation. Marxist theories, often informed by indigenous voices like José Carlos Mariátegui, were used to substantiate theological thinking in these early days, but Christian and theological identities remained central and were persistently defended. That is, liberation theology was concerned about theological positioning, expression of incarnate faith, with denunciation of oppression and sociopolitical motivation. Marxist theories were part of the theological debate not because of some obscure academic interest, but because these were broad popular conversations, linked to indigenous Marxisms and organizing on the Left. Leonardo and Clodovis Boff, Julio de Santa Ana, Enrique Dussel, and liberation theology's parents Gustavo Gutiérrez and Rubem Alves, stand out in this first generation.

Years later, with the end of dictatorships and the emergence of even more consolidated economic oppression through external national debts—disastrous social effects caused by industrialization—theologians such as Hugo Assmann, Franz Hinkelammert, and Jung Mo Sung began to argue that the option for the poor, which continually demands the contextualization of the experience of oppression and suffering, must not lose sight of the implicit agreements and common structures of dominant theology and economics. This new theological generation marked another milestone because it read the "signs of the times" (Vatican II) in new ways, noticing that the economy was a culture, a way of defining how to live and die, rather than merely the subject of an exact science.

In other words, based on classical texts developed by Walter Benjamin[5] and Max Weber,[6] which demonstrate the perverse changes of capitalism, the third generation of liberation theology takes a critical leap, realizing that it is not enough to denounce the situation of the poor of the so-called "third world," thrown into oppressive misery in search of liberation. It is necessary also to understand, in light of theology, how the structures of domination and oppression are sustained, classified as saving, and established as good news. This moment is perceived by Assmann and Hinkelammert in 1989:

> The themes we address most closely are the way economic rationality "hijacked" and functionalized essential aspects of Christianity; the way "economic religion" triggered a massive process of idolatry, which finds its expression more evident in the supposed self-regulation of market mechanisms; and the way economic idolatry feeds on a sacrificial ideology that implies constant sacrifices of human lives. We speak directly of theological notions present in the economy. In other words, it is claimed that economists are also, in their own way, eminent and dangerous theologians. But what interests us is not exactly to launch *accusations, but to reflect on the implications this has for the direction of economic policies and for human problems in general*. (Assmann and Hinkelammert 1989, pp. 9–10, our translation)

These references to economic policies and the human problems lead to the understanding that modern economic rationality is based on mythical foundations, which, in addition to not considering the real life of the population, are based on schemes that demand the sacrifice of human lives. This ethical-theological denunciation opens new paths of theological production, but the most important of all is that it reinforces that real life, mundane and experienced, should be the starting point and ultimate criterion of any elaboration, be it of an economic or theological nature, with the goal to improve the way people produce and access means of survival. From this moment on, liberation theology becomes more than a methodological, ethical content; it is now also a continuous source of revelation of idolatrous structures that victimize anything that promotes life. We will come back to this in the following section.

### 2.2. Idolatry, Sacrifice, Victims, and Other Intersectionalities: New Categories of Approach to an Old Oppression System

In opposition to classical theological perspectives, and entering the discussion of the materiality of life and the human body, which has basic needs such as food and shelter, Rubem Alves indicates the following:

> The liberation of the human being has nothing to do with the denial of the body, but with his liberation from everything that represses it, which does not leave it free for the world or the free world for him [ ... ] The Messiah, the power of liberating freedom, is 'flesh'. There is no place for a God who gives himself to man or who works outside the material conditions of life [ ... ] God is found among the things he gives humans. (Alves 1987, pp. 204–5)

In this perspective on the incarnation, promoted by the first generation of liberation theology, theologians turned their attention to the material and physical experience of people. The body, once perceived as a deposit of sin, would come to be contemplated as a gift that should be treated with dignity. The body is the first "space" where we show solidarity, where we perceive ourselves as a *self* in contact with the *other*.

However, it was necessary to understand how certain inversions originating in the economic universe changed the understanding of the body and of the human being itself. From the 1980s onwards, the so-called DEI School (Departamento Ecuménico de Investigaciones in Costa Rica) pioneered this issue by indicating that the capitalist free market, a human institution, had gradually become a god—actually, an idol. Assmann and Hinkelammert, the key thinkers of this school, pointed out that "idols are the gods of oppression" (Assmann and Hinkelammert 1989, p. 13), representing an inverted image of divinity, constantly requiring the sacrifice of the poor, while, on the other hand, promising life. The capitalist market, made an idol, pretends that people do not have basic needs but only desires that can be fed into a constant economic-sacrificial circuit (Sung 2007). The theological-economic critique of the DEI school, and of the entire generation of liberation theology they represent, can be summarized as follows:

> On an international scale, despite (and perhaps because of) the huge accumulation of wealth in rich countries, the socioeconomic problems of most of humanity are looming and sharpening; and the dominant economic science, deeply committed to established interests, being unable to dismember itself from god dogmas, works in favor of the perverse existing structures and does not admit alternatives that depart from them. In theological terms, this means that there is a solidly established idolatry, and that worshipped deities do not favor the creation of gospels (good news) for humanity. Established gods are hardened gods, especially when they originate from a long and difficult previous metamorphosis [ ... ]. Concrete certainties about real hunger, real death, and all real needs have disappeared. They can no longer be known and determined, because these economists only know beings-with-desires who, apparently, *have astral bodies*. Then, *all concrete demands become debatable*, nothing can be known for sure, everything is unlimitedly complex, and nothing preserves the simplicity of tears, cries, hunger, and the danger of death. (Assmann and Hinkelammert 1989, p. 36, our translation)

Faced with the naturalization of people treated as sovereign consumers and not with dignity (Sung 2020),[7] outside the key of needs and within the key of desires, what we classify as sacrifices are understood as viable, legitimate actions and unique possibilities of achieving seemingly positive results. Sacrifice must be hidden so that it continues to produce meaning and cannot be denounced or brought to light in how it perpetuates itself on the basis of capitalism. Life is promised through death, peace is promised through violence, and abundance is promised in the midst of lack. And so we are led to believe that material life, the dignity of the human being, is more a commodity than anything else. In other words, what this third generation of liberation theology does is the production

of a theology that denounces oppression, reveals the idolatry of the market, and indicates that its result is the sacrifice of human lives. The target is a methodological and practical strategy that defines the meaning of life and turns it into a search for satisfaction through the market.

In this sense, as the Chilean theologian Pablo Richard warned, it is necessary not only to identify the existence of idols, to denounce them, but to constantly promote the "anti-idolatric discernment of false gods, fetishes that kill and their mortal religious weapons" (Richard 1982, p. 7). From this idea of discerning, we understand that, although the option for the poor is the classical methodological route of liberation theology, it is not sufficient to denounce specific oppressions that accumulate in various formats around the world. In fact, since the 1990s, some Latin American liberation theologians saw in the classical option for the poor a non-liberating scope. That is, from this time on it was argued that it is not enough to produce theology looking at macroeconomic contexts; one must also look at the deepest examples of everyday life, where oppressions and sacrifices are even more cruel. In other words, it takes more than talking about the poor; instead, it is necessary to develop more appropriate understandings of the problems and the potential of the people together with the people. This includes more adequate class analysis that goes deeper into exploitative relationships.

In this context, we highlight the work of theologian Ivone Gebara, a Catholic nun, who, when being silenced by the Vatican produced a thesis on the problem of evil, focusing on the experience of Latin American women, who were and remain the most sacrificed of all. For her, it was not enough to use the category of the poor as an epistemological-ethical foundation, it was also necessary to talk about how poverty painfully affects the lives of women, especially when associated with other types of oppression. Poverty, analyzed in a transversal way by Gebara, represents not only an economic situation, but an overlap of genders, a religious (Christian) morality, and a racist practice. This critique was and remains pertinent because the so-called option for the poor lacked analyzing and acting together with the poor themselves. Without names, addresses, or voices, liberation theology denounced a sacrificial structure that is latent to this day, but remained on the surface when the subject is to enter the reality of the most ordinary life (Das 2007). That is, there was talk about real people, but without sufficient qualitative approximation.

In this sense, a step forward, understood as a future for liberation theology, is to use a category that encompasses a larger portion of the capitalist operations that we have dealt with so far (idolatry, sacrifice, concealment, etc.), which deepens social and theological analyses and pushes beyond simplified perspectives. We would suggest, therefore, the use of the category of the *victimized* rather than merely of the *poor*, because it is necessary to highlight that in addition to unfavorable economic situations there are other mechanisms that limit the lives of people. As a result, only a very small group of people are able to accumulate and dominate. Historically in liberation theology, the term *poor* was almost always accompanied by the auxiliary expression *preferential option*, which indicates the central methodological option for Latin American liberation theology. What cannot be lost sight of is that this way of doing theology not only presented another possible image of God but was concerned with a way of being in the world marked by inequality and exploitation. That is, liberation theology, by making the poor the center of its production, denounces the fact that real life was/is marked by a dehumanizing poverty, because of various historical processes and structures.

When we prefer to use the category 'victims' instead of 'poor', we do so (1) in an attempt to go beyond an analysis of reality that might be too generic because poverty is not homogeneous; rather, it is variable and needs to be observed in each context and time;[8] (2) because the term 'victim' denotes that precariousness and exploitation have become worse and has deepened with the evolution of capitalism in its current neoliberal forms. That is, poverty (and here we include deaths caused by poverty as well) is not an accident, it is a necessary condition of capitalism and therefore constructed by it; and, finally, (3) because the term victim promotes a transversal discussion (including the categories 'race' and

'gender', in addition to 'class'), pointing out that poverty, exploitation, and inequality have other layers, depending on the group of people who experience them. In the case of Brazil, for example, black women and LGBTQIA+ people are the most exploited and find themselves in situations of greater social vulnerability. In the case of the United States, it is people of BIPOC and queer communities. In short, the victim category not only reveals the complex problem of poverty, it also observes it as a result of economic-social systems crossed by diverse forms of systemic violence, such as racism, sexism, and the exploitation of the labor force.

In sum, the category of the victimized helps to reveal how people are constantly exploited by capitalism and related structures. Talking about victims requires a more complex analysis as not all victims are the same, but it also reminds us of the structural roots of victimization. And not all who claim to be victims belong into this category; for instance, oppressors who misinterpret being challenged as being victimized. For the discourse on poverty and the poor, this means that the poor do not cause their own poverty and it requires a protest of the conditions that victimize them, as well as working towards alternatives. For the most part, the poor are victims of economic processes and of social evils that fragment relations and turn both human and other-than-human nature into commodities.

Producing critiques and theological content from the category of the victimized not only reveals a system that reverses life and death but also makes room for people themselves to participate, speaking of their experiences, and recognizing privileges and lack thereof. Ultimately, within the neoliberal capitalist logic, the proverbial 99 percent that have to work for a living experience some form of unavoidable exploitation and lack of power/dignity (Rieger 2022). For an even more liberating future, it is necessary to recognize and denounce the situation of specific groups within the 99 percent, including poor black women, stigmatized poor black men, queer people, whites descended from poor immigrants, young white women, the indigenous elderly, and many other realities where sacrifice is the norm. In addition, ecological devastation affects rich and poor, millionaires and the miserable, including everyone in what might be understood as catastrophes (Rieger 2022; Day 2016; Boff and Moltmann 2014).

In a recently published article, Priscila Silva and Sung (2022) discuss how it is necessary to listen to people who are in less dignified situations in order to understand the many dimensions of sacrifice and idolatry, including the co-optation of theology and the church for capitalist purposes. In this article written in the spirit of liberation theology, the socioeconomic structure of Latin America or of historically oppressed countries is not at the center. Instead, the testimony of a black woman, who at the height of the pandemic confesses her faith and hope in God rather than in the system, the state, or social policies, is emphasized. What this move indicates is that, in addition to conventional categories and methodologies, it is necessary to aggregate experiences and grievances that directly begin with those who suffer the most harmful effects of the socioeconomic system. In order to conceive of a fruitful, critical, and ethical trajectory for liberation theology, we should ask ourselves, 'What are the categories that the poor themselves use to classify their lives?' We might be surprised by alternative ways of understanding this complex sacrificial system, which is neoliberal capitalism.

This movement of approximation to ordinary life, from the broader use of the category of the "victimized" to the understanding of multiple experiences in the midst of neoliberalism, dates back to the beginning of liberation theology, the ecclesial base communities (CEBs), and the popular reading of the Bible, which was the foundation of the theological production of liberation. Even before liberation theology became an academic discipline, it was popular, pragmatic rather than analytical, both simple and complex. Even the connection to the social sciences came later. Gebara (2020) indicates that, because it has distanced herself from people's daily experience, liberation theology has become more an object of research rather than an ethical-methodological theology. With the weakening of base communities, given constant clerical/institutional pressures, it seems that liberation

theology has lost its largest field of popular cultivation, and the conception of its role as the protagonist of the poor has entered into crisis. That is, much of the founding structure of dialogue has been lost over the decades.

This situation, in a way, persists at present and therefore there is a need not only to defend the theological method born at the end of the 1970s, but to redevelop it from the cries of those who suffer the most unfair treatments. Unfortunately, scholars often fail to grasp this challenge because they are distant from what is going on every day. Nevertheless, it is important to note that there are people, projects, and perspectives attentive to this issue and people who, therefore, already operate theologically and sociologically from the experience of what might be called the "factory floor". To illustrate, we will now show how the perspective of solidarity can be a new ethical-theological key, critical of the strongest Christian pillars that have been co-opted by language and economic culture, such as common models of charity. This topic will indicate, above all, how it might still be possible to develop liberation theology and take steps that are historical signs that liberation is not primarily utopia but concretely happening even if the empire persists. In order to gain freedom, in this perspective, it is necessary to dialogue with and organize people, and to reclaim alternative forms of power.

### 2.3. Experiences of Liberation in Daily Life: Organizing Solidarity Circles in the United States

The so-called "historical project" of liberation theology has undergone shifts and transformations over the past five decades. While early on real alternatives to capitalism seemed possible, things have changed with the onsets of neoliberal capitalism, military dictatorships, the end of Soviet-style communism, structural adjustment policies, and challenges from the political Right in various countries (Donald Trump and his followers in the United States, Jair Bolsonaro in Brazil, etc.). In some cases, sheer survival is now the main goal, especially when substantial parts of the population are living below the poverty line. This is, of course, most urgent in many places in the Global South, but at the height of the COVID-19 pandemic in the United States, 29.5 percent of families were food-insecure (Silva 2020).

When times are difficult, organizing becomes more important than ever. Offering critique, protesting, and other popular forms of resistance are not enough. Even individual struggles for sheer survival can benefit from being organized, as they depend on building community power and relationships. Understanding the prophetic traditions as mainly "speaking truth to power," as they have often been interpreted, is inadequate if the prophets have the truth but the dominant system still has the power. Perhaps the most common but also the most problematic misunderstanding of liberation movements is that they are mainly protest movements—speaking out against the dominant system, offering critiques and making moral demands on the dominant powers, trying to change policies here and there without building alternative power and changing systems (political, economic, and religious).

In the United States, some of this is related to myopic interpretations of the Civil Rights movement that neglect the deep organizing and power-building that went on. The 1963 March on Washington, for instance, was about "Jobs and Freedom," rather than merely about civil rights, and it was put together by people like A. Philip Randolph and Bayard Rustin, closely linked to the labor movement, concerns for economic power, and the everyday life of African American communities. Instead of playing off matters of race and class, Randolph, Rustin, and many of their collaborators sought to fight the two evils in relation to each other. They knew what is often forgotten today—namely, that building economic power in minority communities can do more to fight racism than many other efforts, and dealing with racial divisions at the workplace by organizing workers can do more to build the power of working people than conversations about poverty or analyzing class structures can do by itself. It is perhaps not surprising that both Martin Luther King, Jr., and W.E.B. DuBois noted that the labor unions were the most effective organizations involved in the struggle against racism (King 2011).[9]

All that is to suggest that popular struggles for survival, instead of limiting themselves to protesting and resistance, have often understood that something needs to be built—the power of working people, in the example of the Civil Rights movement. It is hardly a coincidence that King was assassinated when he helped organize the sanitation workers in Memphis. In the United States, this constructive approach can be seen also in a long history of African American efforts to build solidarity economies and worker cooperatives (Nembhard 2014). Even the Black Lives Matter movement in the United States, often perceived as a mere protest and resistance movement, has developed an economic platform that puts black liberation on the footing of building economic power but is unfortunately not very well known.[10] In Latin America, the ecclesial base communities of the early days of liberation theology offered support networks where new relationships were established and community power was rooted. Today, the Movimento dos Trabalhadoras Rurais Sem Terra (MST) in Brazil pursues constructive projects that lead to building alternative relationships and power: going beyond conversations about land and landownership—fashionable in the United States at present—land here matters because it is part of the production of life.

These examples inspire some new efforts to bring together faith communities and projects in the solidarity economy, guided by the ever-evolving conversations of liberation theology. The Solidarity Circles of the Wendland-Cook Program in Religion and Justice at Vanderbilt University provide a case in point. Putting together faith commitments and economic agency—we are also talking about religious and economic democracy—is not just a matter of economics: at stake is an approach that engages all of life and feeds back into the development of faith. Moreover, the alternative power that is built here addresses not only inequalities along the lines of class but also along the lines of race, ethnicity, gender, sexuality, disability, and even age, because all of these aid the relentless exploitation of the capitalist system (BIPOC, women, non-gender-conforming people, differently-abled people, and older people generally get paid less, has less power and experience greater pressures at work).

Each solidarity circle is a virtual peer network that brings together about a dozen representatives from faith communities to "investigate, educate, and organize".[11] Each individual faith community engages in a specific project of the solidarity economy in consultation with seasoned organizers, including worker coop developers and labor leaders, in conversation with theologians and religious ethicists. These solidarity circles are designed to harness the mutual connections of economic and religious developments: in a world where all of life is dominated by the forces of what is sometimes called the "Capitalocene,"[12] developing alternative economic relationships determined by the interests of working people rather than employers or stockholders creates both power and freedom that are mostly absent today; these relationships, in turn, can inspire alternative religious relations determined by the faithful themselves rather than hierarchical structures that often resemble the structures of corporations. Religious inspiration and theological study are closely connected with these dynamics, based on the observation that God is often at work in the world most intensely where the pressure is greatest (back to the preferential option for the poor and not the victims), and that this is the context in which studies of biblical and other traditional sources of respective faith traditions is most fruitful. All of these components are processed over nine months in terms of the best insights of broad-based community organizing, aided by educational resources and monthly group meetings facilitated by theologians, ethicists, and some of their students.

The fundamental question underlying these efforts is how to build and construct the kinds of projects that might present real alternatives to the dominant status quo. In our experience, minority projects and even minority movements can always be co-opted by the status quo in terms of what is now called "inclusion". Yet the status quo of neoliberal capitalism is hardly challenged or transformed by the inclusion of a few leaders from minority groups. To the contrary, the common mantras of diversity, equity, and inclusion are welcome opportunities to expand the reach of capitalist domination and put it on broader shoulders without ultimately benefiting the majority of minorities.

Real alternatives to the dominant status quo require some form of solidarity among those most affected. Solidarity economies are places where such solidarity can be embodied and practiced. What we have called "deep solidarity" (Rieger 2022) is located here, as solidarity economies bring together a variety of people from all walks of life. Unlike right-wing solidarity, which is based on sameness and identities (usually misleading ones, such as race, gender, and nationality that mislead working people to assume they have more in common with their white, male, or American bosses than with non-white, female, or non-American workers), deep solidarity is based on practical collaborations where it matters most: in producing alternatives to exploitation and in the production of wealth for the benefit of the community. The fundamental paradox of deep solidarity is that it is does not require sameness but becomes stronger the more diverse those who collaborate are. This is true not only for racial, ethnic, gendered, or sexual identities, but for different religions too. Inter-religious dialogue takes on totally new forms here (Rieger 2021), and entirely different relationships can be built and strengthened. The goal is not to become more alike, to erase difference, or even to "see" and "hear" others—the goal is to work together to build a different world, starting with the places where exploitation is most severe and damaging: at work, where most people spend the largest part of their waking hours; or if they are excluded from work as an increasing number of people are, in informal economic relationships that are often profoundly dehumanizing and even destructive.

Deep solidarity is international by design, understanding that those who are exploited in the Global South are connected to those who are exploited in the Global North—making up the majority of the population in each place. Unfortunately, this has often been covered up by the misuse of certain theories, including the classical theory of dependence, which some have taken to mean that people in the North are generally wealthy and people in the South are not, preventing a deeper sense of solidarity among working people from the outset. The only kind of solidarity that can be imagined in this context is one according to which those who are more privileged put themselves on the side of those who are less privileged, making solidarity a moral matter that can be engaged as quickly as it can be abandoned. A distinction of privilege and power might help reconceptualize what is at stake and deepen the notion of solidarity.[13]

Privilege, of course, is real and can be observed in many places: being a resident of the Global North carries certain privileges, as does being white, male, heterosexual, and so on. Privilege also accrues to certain national, professional, and religious identities. However, privilege does not necessarily translate into power, especially the power to change things. The confusion of privilege and power is very useful for those who seek to preserve dominant power. North/South divisions may serve as an example: US privilege is actively referenced to suggest to workers in the United States that they have more in common with their American employers than with workers elsewhere. This approach is often used by union busters, and political and religious forces on the right. This leads to the commonly observed phenomenon that working people often vote against their own interests as well as believe religious and other doctrines that go against their interests. The result is frustration all around, because US workers are not benefiting from this confusion of privilege and power, and international solidarity is undercut because workers elsewhere are left unsupported. A distinction of privilege and power can help clarify things: according to a rule accepted in most capitalist economies, corporations exist for the benefit of their stockholders (and to some degree of their consumers), but never for the benefit of their workers. In the United States, Henry Ford was sued because some of his stockholders believed he did not sufficiently pursue the interest of his stockholders.[14] Power, therefore, lies in the hands of the stockholders (not the average US citizen who may hold a few stocks but does not control large amounts).

One of the most important insights, necessary to broaden the work of liberation theologies today, is to realize that dominant power is shared by relatively few people—the proverbial 1 percent, which is more likely the 0.1 percent. This means that the power of all others is limited, and even those who assume that their privilege translates into

power—professionals, pastors, professors, middle managers, most politicians—need to take another look at what is going on. Solidarity emerges when people realize that they are not benefiting from the dominant powers as much as they think, which includes an increasing number of members of the middle class, whose fortunes are dwindling. Once this is clear, whatever privilege people have can be put to use for the causes of liberation from dominant power, resulting in a deconstruction of privilege.

This conversation further broadens the classical notion of the preferential option for the poor and the victims, but it also brings it back in a stronger form that is less based on morality and more grounded in reality. If the suffering of some is connected to the suffering of all, as the apostle Paul argued in 1 Cor. 12:26, the realities of exploitation, extraction, and oppression deserve our utmost attention. More specifically, those who experience various interlocking forms of exploitation and oppression in their own bodies, via the intersections of class, race, ethnicity, gender, sexuality, and ability, are the ones at the heart of the alternatives that are being built. Those who are more privileged need to pay attention to those with less privilege, not in order to establish another Olympics of oppression, but to understand what we are up against and what it takes to liberate ourselves collectively. The theological part of this process has to do with an understanding that this is where God is found and at work—reclaiming the old insight of liberation theology that there is an option for the poor, not because the poor are good but because God is good.

## 3. Conclusions

More than fifty years after the emergence of liberation theologies and other contextual theologies that aimed and still aim at the establishment of a more just world, we find ourselves in a period of history where the levels of inequality and the exploitation of human beings and other-than-human nature must be considered nothing less than catastrophic. We are witnesses of a time of unprecedented exploitation, in which the largest portion of society leads automated lives in search for survival and experiences only commercialized dignity and the precariousness of human relations, which include the precariousness of work, spirituality, health, and other fundamental elements of daily life.

In a way, after all that we have described here, it can be said that exploitation is not an isolated element of the relation of production in the current capitalist scenario but has become an ontological situation, a condition of life that seems inevitable for most of the population. As many as 99 percent—those who have to work for a living—are forced to live under this reality. Within these 99 percent, it seems that not only are many born poor and die poor, they are also born poor and, throughout life, the mechanisms of exploitation are perfected in such a way that they are silenced, worn out, and rendered helpless by certain images of God, which have more to do with the free market than with the life of the Son, always critical of the most powerful and their empires.

In this brief article we realize that the future of liberation theology passes not only through structural analyses, which are extremely necessary, but through the realization that the construction of power is part of the changes that we want not only to see in the world, but to enjoy. That is, more than pursuing contextual analyses, it is necessary for people to identify their places in society critically, seeing that capitalism is not only exploitative but, in many cases, exclusionary, harming even those who feel to be an unquestionable part of it. And in this lies the difference between privilege and power. Many privileged people mistake privilege for power. The future of liberation theology depends on the distinction of privilege and power in neoliberal capitalism; as the 99 percent realize the substantial limitations of their power to change the system, they can begin to deal with different levels of privilege that can help build solidarity and alternative powers that can help equalize relations and not verticalize them more.

In this sense, theology is a tool of critique, of motivation and, above all, of solidarity, because without a new look at others and the world, we only resign ourselves to accepting what seems normal—the common, the routine, and even destiny (God's desire). In neoliberal capitalism marked by selfishness, solidarity is a political-theological position

that positively reclaims self interest and communal interests. For us, solidarity also has to do with the recovery of a popular hermeneutic of the Bible, of a reading and practice of Christianity that is not disconnected from material life, from suffering and the anguish of living under the constant threat to survival. The future of liberation theologies, in this sense, depends on an ever renewed, committed, and indignant gaze at the dynamic structures that increasingly separate us from one another and from a God who laments misery, and from a more dignified life that will not become a reality without critical analysis, without struggle, and without organizing. While the option for the poor is still a valid methodological horizon, it is necessary to move forward, including at the heart of the project of solidarity the most victimized people who experience constant exploitation, in a search for developing the power that is denied to them. The point is popular participation, which is to say that liberation means building people power. Solidarity is a tool for this, and through this renewal of the mind and of life, more dignity has already been experienced in both Brazil and the United States.

**Author Contributions:** We declare that the all the steps of conceptualization, methodology, investigation, writing-original drafting preparation, writing-review and editing, visualization were done in collaboration by both authors. All authors have read and agreed to the published version of the manuscript.

**Funding:** This research received no external funding.

**Data Availability Statement:** This article is a fragment of new studies in development, including a thesis, and as result of observations in the working field. Both authors are committed to publish new data soon.

**Conflicts of Interest:** The authors declare no conflict of interest.

## Notes

[1]  The list is long, but in the Brazilian context alone it includes prominent figures like Rubem Alves, Leonardo Boff, and Ivone Gebara. Alves was accused in 1968 of being subversive (considered a crime at the time by the national legal system) by the members of members of his own Presbyterian church, claiming that he held some heretical ideas that could cause social and political instability. Boff was silenced by the Vatican in 1985. In 1995, Gebara received the same punishment for denouncing the patriarchy of the church, the sexism in the biblical interpretations, and for speaking about the reproductive rights of poor women in the Northeast of Brazil.

[2]  To understand the deep connections between religion and the capitalism (and its changes), see "No Rising Tide: Theology, Economics, and the Future" (Rieger 2009), and "Desire, Market and Religion: Reclaiming Liberation Theology" (Sung 2007).

[3]  Thinking about the pain caused by this history of dictatorial violence in LA, we highlight a beautiful work of the Argentine theologian Néstor Míguez, formed by ISEDET, in "Um Jesus popular: para uma cristologia narrativa," (Míguez 2013). Writing in a liberating perspective, Míguez composes a possible prayer of a "Mother of the Plaza de Mayo" who had her son disappeared or killed by the Argentinean dictatorship. Míguez continues to produce a theology that is critical, especially to oppressive economic systems.

[4]  To see more about that, read *Open Veins of Latin America* (Galeano [1973] 1997).

[5]  In portuguese translated as *Capitalismo como religião* (Benjamin 2010), "Capitalism as religion".

[6]  *Protestant Ethic and the Spirit of Capitalism* (Weber 2001).

[7]  For more on this topic, consult: https://www.metodista.br/revistas/revistas-metodista/index.php/ER/article/view/10502 (accessed on 9 February 2023).

[8]  Already in his 1998 book *Remember the Poor* (Rieger 1998), Joerg Rieger coined as one of the epigraphs the sentence "*The* poor do not exist".

[9]  This is not to say, of course, that racism therefore is not a problem in the labor movement.

[10]  https://m4bl.org/policy-platforms/economic-justice/ (accessed on 7 July 2023).

[11]  https://www.religionandjustice.org/solidarity-circles (accessed on 7 July 2023).

[12]  The term is conceived in contradistinction to the claim that we live in the so-called "Anthropocene," where humanity as a whole has taken over the fate of the planet. See *Theology in the Capitalocene* (Rieger 2022).

[13]  For the distinction of privilege and power see *Theology in the Capitalocene*, chapter 4 (Rieger 2022).

[14]  See "Dodge v. Ford: What Happened and Why?" https://corpgov.law.harvard.edu/2021/12/01/dodge-v-ford-what-happened-and-why/ (accessed on 7 July 2023).

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
