# Peer review of "Liberation Theologies and Their Future: Rethinking Categories and Popular Participation in Liberation"

_religions, doi:10.3390/rel14070925_

Round 1
Reviewer 1 Report
In line 37 on page 1, the authors state that liberation theologians were accused of being traitors and rioters. A reference indicating which liberation theologians were accused of this, by who and why would be helpful here.
Author Response
We will answer this question in the text.
Reviewer 2 Report
I very much appreciated the spirit and direction of this article in charting a course for liberation theologies to go forward in addressing contemporary forms of social and political oppression by way of consensus and coalition building. However, I noted several points at which revisions seemed to me necessary. I will note each of these in order, referring to the manuscript by line number. Then I’ll observe some problematic patterns I see among the points I’ve raised, identifying those that represent the most significant barriers to publication.
- The definition of empire as “totality that generates death” (39-40) is inadequate: it entails that malaria or the surface of the sun are imperial realities. The generation of death is necessary but not sufficient for marking out empire.
- There seemed a tension between identifying liberation theology with a popular movement dating back to the 50’s rather than as the intellectual production of Gustavo Gutierrez (54-57), btu then you later go on to define liberation theology primarily with reference to GG rather than anything about the movement you mention (121ff)
- Authors claim that there are “constant discussions” of whether liberation theologies still make sense (66-67) but we seem to need a citation and evidence. Who actually says this?
- Authors should clarify that they are referring to the global north/south rather than U.S. North/South (88-89)
- The outline of engaging capitalism, its idolatry/sacrifice, and commending solidarity circles seems clear enough, but what is the thesis? It is unclear exactly what the author’s overarching aim is or to whom it is addressed or what motivates it. It seems likely to be read as an edifying discourse or sermon of sorts to those particular liberation theologians who already accept a particular characterization of the relationship between economic justice and liberation theology. But why is such a sermon needed? If on the other hand it is supposed to be for those who don’t see the need for a liberation theology, who are those people and how might they be represented in the text?
- The claims about classical Christianity producing secularization are contested, citation of whose arguments the author accepts seems necessary (Casanova? Taylor? Gregory? Pfau?) (106-111)
- A big problem seemed to me to be the paper’s referring to three “generations” of liberation theology, offering a kind of periodized taxonomy of liberation theology (149). The trouble is that the author fails to clearly mark out what distinguishes these three generations from one another, or what sort of actual development justifies the periodization. At 149 we have a characterization of first two gens, but that pgph concludes as summary of only the first gen (160), before going on to a discussion of a third gen (171ff). So we seem to lack a characterization of a 2nd gen and what marks it out from first and third. Moreover, at 210 we are introduced to the theme of capitalist idolatry as a product of first gen, but then the authors go on to name figures identified with 3rd gen. 251-253: so idolatry and sacrifice have been identified with both first gen and third as if it were a distinctive feature of the latter.
- At 262-3: we are given this internal critique among liberation theologians regarding the inadequacy of the option for the poor. The idea seems to be that this notion is too abstract or coarse-grained as a macroscopic economic analysis in order to capture the particularities of oppression experienced in everyday life. If this critique is characteristic of a larger scale movement in liberation theology, much more citation is needed. But a further problem is that to illustrate the point the authors appeal at 274 to Gebara’s focus on women’s lives. But this does not indicate the overly general nature of option for poor but rather the irreducibility of social oppression to matters of economic disenfranchisement, whether the analysis is at the macro or micro scale. Gebara seems to point to a the intersectional nature of oppressions that makes an economic analysis too reductive, rather than being merely too general or abstract 281-2
- The problem of economic reductivism is hinted at in 288 with “other mechanisms” talk but not elaborated along the lines of any intersectional analysis. The idea that the problem of economically reductive analyses can be addressed semantically by preferring “victim” to “poor” seems inadequate because the primary explanatory mechanism still bottoms out in “a system” (295-6) of “neoliberal capitalism” (319). For this reason, the notion of “aggregating experiences” (314-15) seems contradictory to the diagnosis offered at 281-2: the authors have (mistakenly it seems to me) mistaken the problem of economic reductivism for a problem of over-generality or abstraction, but here they propose remedying it by moving exactly in the direction of aggregating all particular oppressions into their economic common denominators. But this lands us back in “the poor” just with a different name i.e. the “victim” of neoliberal economic disenfranchisement. For this reason the idea of intersecting oppressions at the end (503-4) seems to cut against the authors’ own reductionist commitments, or else the appeal to many irreducible oppressions with class being one along multiple others is misleading or illusory
- “In some cases, sheer survival is now the main goal” (354-55)? Hasn’t liberation theology always been framed as a matter of survival as a pre-requisite for some more enriched notion of human flourishing? Hard to see what is novel here.
- At 428-31 the authors name a real problem of co-opting “inclusion” into capitalist reproduction as opportunity to expand reach of its domination (434)., but worker co-ops may likewise consistently reproduce and reinforce racist, sexist, and ableist oppressions even while consistently seeking economic justice for people of color, women, and the disabled. The trouble of a purely economic analysis of liberation/oppression is just how to address this situation, and one symptom of that trouble is the tendency to contrast mutual understanding and the goal of building “a different [economic] world” (451-2)
- The distinction between privilege and power (471) seems to be an important point, but needs greater elaboration, including further corroboration or citation for the claim that workers vote against their interests because they internalize this conflation of privilege and power and find more solidarity qua American with their bosses than with international workers (473ff).
From these worries about the paper, I detect a few overarching problems that would require remediation before I could commend this article for publication. First, it would need to be framed with a much more clearly stated thesis that is motivated by some problem or question that is more clearly marked out. This might help to limit the scope of the claims being made about the future of “liberation theologies” when the paper in fact only seems to address Latin American liberation theologies and neglects the flowering of all manner of other liberation theologies with a diverse array of methodological commitments, rather than being more classically Marxist in character. Second, the paper would need to take more seriously the contested status of fundamentally material and economic rather than cultural and identity-grounded analyses of the oppression to which liberation theologies are ordered. Finally, the paper would need to include more attention to the scholarly literature that underlies the claims being made throughout the paper, especially where the point is to simply reference the status or role of those claims rather than seeking to offer argument or evidence for them.
Author Response
We are appreciative to this reviewer for the detailed review. Regarding the scope, we will be making it clearer that we are talking about generations of liberation theology in relation to the Latin American context. What we are trying to show there is that the broadening of the topic of liberation is not primarily a matter of adding in other types of liberation theologies but grows out of developments in Latin American itself. This includes the tension the reviewer sees between a “Marxist” focus on economic realities and another focus on the “diverse array of methodological commitments.” In our view, this tension is not as pronounced as it often appears, as even in the beginning the focus on economic issue was never purely “Marxist” and certainly never determinist, and the focus on other methods grew organically from the struggles of the people, which from the very beginning had to negotiate race, gender, ethnicity, etc. In other words, the contrast between economic and cultural analysis is not as severe as often claimed, and it is constantly shifting. This is part of the overall argument of the article. Finally, we’re very appreciative of the reviewer’s reference to the privilege/power distinction made in the article. The literature referenced responds to the suggestion to explain this distinction more in-depth, especially since one of the authors has been part of the foundational development of this discourse.
Reviewer 3 Report
This is an interesting and very valuable contribution to the ongoing debate around liberation theology and its future, with particular reference to work in Brazil and work with an economic focus and a critique of idolatry. It is a pleasure to read and balances sophisticated analysis with an accessible writing style. As will be clear from the above I believe it is a strong piece and the comments below are no more than minor suggestions which might strengthen it further.
1. The suggestion that the language of 'victim' might be preferable to 'the poor' is of particular interest. This language of victim has particular resonance with 'the crucified people' as developed by Ellacuria and Sobrino and this might be noted as a further element in its favour. However, in discussions of sexual abuse the language of 'victim' is usually avoided in preference to the language of 'survivor', as 'victim' is associated with a loss of agency. Might a similar critique apply here (or not?)? The language of 'the oppressed', 'the excluded' or 'the marginalised' might also be considered, and could also be noted as possible terms.
2. Many of the references to writings by liberation theologians are to the Portuguese editions. This is welcome (and appropriate, even when some were first published in Spanish) but some of the dates may be misleading. For example Gutierrez 2000 is to the 9th end of a book that was originally published in Spanish in 1971. The authors themselves are doubtless fully aware of this, and many readers will also be aware, and there is usually a good reason for brevity and simplicity in notes, and a house style that favours more up-to-date references. However, given the chronological development of ideas is important within the article (references are made to first generation and second generation) adding first publication dates in square brackets for these works in the bibliography would be a welcome addition and is likely to be helpful to readers who are less familiar with chronological developments within liberation theology.
3. A few of the references might be double-checked. For example, for Sung 2007, the publication place given is 'Northernwestern' when SCM Press is normally cited as London. Likewise, for Galeano 1996 no place is given, when Monthly Review Press is normally New York. For Galeano 1996 it might also be worth noting that the original publication was 1973.
Thank you for the chance to read this work.
Author Response
Thank you for your excellent suggestions. We have made substantial revisions in response to point 1, explaining in more detail why we use the language of “victim.” We have also made changes in response to comments 2 and 3, making suggestions for the bibliography which were easily fixed.
Round 2
Reviewer 2 Report
I find all of my critical remarks to have been adequately addressed in this revision. It is a useful article and I thank the author(s) for their care in making the appropriate revisions.